# Changes in Vertebrobasilar Artery Dissection Visible with High-Resolution Vessel Wall Imaging: A Serial Follow-Up Study

**DOI:** 10.3390/diagnostics13233585

**Published:** 2023-12-01

**Authors:** Eunjeong Cho, Youjin Won, Ui Yun Lee, Seung Bae Hwang, Hyo Sung Kwak

**Affiliations:** 1Jeonbuk National University Medical School, Jeonju 54907, Republic of Korea; dmswjd9224@jbnu.ac.kr (E.C.); youjinwon78@gmail.com (Y.W.); 2Division of Mechanical Design Engineering, College of Engineering, Jeonbuk National University, Jeonju 54896, Republic of Korea; euiyun93@naver.com; 3Department of Radiology, Research Institute of Clinical Medicine, Jeonbuk National University, Biomedical Research Institute of Jeonbuk National University Hospital, Jeonju 54907, Republic of Korea; sbh1010@jbnu.ac.kr

**Keywords:** vertebrobasilar artery, dissection, vessel wall imaging, stroke

## Abstract

Background: High-resolution vessel wall imaging (HR-VWI) can identify vertebrobasilar artery dissections (VBADs) due to its good intramural hematoma and intimal flap visualization. Although the clinical course of VBADs is known to be benign, changes in VBADs visible using HR-VWI at follow-up are unknown. Thus, this study aimed to assess serial changes in VBADs using HR-VWI at follow-up. Materials and methods: Patients with neurological symptoms from VBADs who had undergone both initial and follow-up HR-VWI examinations were retrospectively enrolled. Enrolled patients with VBADs at the initial HR-VWI after acute symptom onset underwent serial follow-up with HR-VWI at 3, 6, 12, and 24 months. Patients were classified into three groups based on the results of follow-up HR-VWI examinations: type 1 = wall thickness of the dissected artery; type 2 = no interval change; and type 3 = occlusion. Results: Fifteen patients (median age: 50 years, nine males) were enrolled in this study. All patients initially showed an intimal flap and a double lumen. Twelve (80%) patients showed strong wall enhancement. Nine (60%) patients had an intramural hematoma. During serial follow-up, nine (60.0%) patients showed type 1 lesions due to attachment of the intimal flap to the vessel wall, five (33.3%) showed type 2, and one showed type 3. Four patients with BA dissection showed type 2 lesions without change in the intimal flap or the double lumen. Conclusions: Changes in VBADs in HR-VWI were observed during the follow-up period. Most patients with VBADs showed the healing process, such as the disappearance of the intimal flap and the double lumen.

## 1. Introduction

The estimated incidence of cervical artery dissection is 2.6 per 100,000 person-years, with vertebral artery dissection accounting for about half of all cases [1]. The incidence is highest among young and middle-aged adults, showing a slight male predominance [2]. Vertebrobasilar artery dissection (VBAD) contributes to subarachnoid hemorrhage and/or stroke [3]. Clinical presentations of VBADs can be classified based on their underlying mechanisms and time course. Transmural dissection presents acutely with stroke or hemorrhage secondary to rapid intramural hematoma formation or rupture [3]. In particular, ischemic symptoms can occur due to occlusion of the intramural thrombus.

Radiological diagnosis of an arterial dissection was made when more than one of the following imaging features was present: (1) intramural hematoma; (2) intimal flap or double lumen sign; (3) dissecting aneurysm with or without diffuse wall enhancement at the culprit vessel; and (4) focal stenosis or string of pearls sign [3,4]. Traditionally, multimodal diagnostic imaging techniques such as digital subtraction angiography (DSA), time-of-flight magnetic resonance angiography (TOF-MRA), and computed tomography angiography can reveal flow patterns and provide information about the vascular lumen. However, these imaging modalities can only evaluate vessel configuration, not the vessel wall [5].

High-resolution vessel wall imaging (HR-VWI) is applicable for differentiating various blood vessel conditions. It is a helpful method for tracking and investigating structural changes within blood vessel walls [6,7]. T1-weighted imaging in 3D-VWI can directly visualize vessel wall intracranial VBAD lesions during the acute stroke period compared with multisequence MRI [8]. It also offers excellent visualization of the lumen and vessel wall to confirm imaging findings of intracranial dissections [9,10]. Jung et al. [10] have reported that HR-MRI can be used to distinguish morphological features of chronic stages of spontaneous and unruptured intracranial dissection (ICAD) as complete normalization, complete normalization with minimal wall changes, incomplete normalization, dissecting aneurysm, and occlusion. Hanim et al. [4] have reported that younger age, stenosis improvement, disappearance of wall enhancement, and intramural hematoma are more frequent in an extracranial dissection than in an intracranial dissection in follow-up HR-MRI findings. However, it is remarkable that data focusing on structural changes to the VBA following a dissection are currently rare. Although the clinical course of VBADs is known to be benign, changes seen using HR-VWI in follow-up studies are not well known. Therefore, this study aimed to assess changes after VBADs by comparing initial and follow-up HR-VWI examinations.

## 2. Materials and Methods

### 2.1. Study Population

This study was conducted with the approval of our local institutional review board. Each patient provided informed consent before their images were obtained. Thirty-three patients who were consecutively diagnosed with VBAD using HR-VWI between January 2018 and January 2023 were selected. Initial routine brain MRI and MRA were performed for all patients to evaluate neurological symptoms. HR-VWI was performed to evaluate VBAD within one week after the initial MR examination.

Inclusion criteria for selecting patients for this study were as follows: (1) acute onset of headache and/or ischemic symptoms related to lesions of a VBA territory; (2) HR-VWI examination within ten days of symptom onset; (3) imaging findings of VBAD from HR-VWI; and (4) patients with planned serial HR-MRI performance at regular intervals during the follow-up period (3 months, 6 months, 12 months, 24 months after the initial MRI examination). Patients who met any of the following conditions were excluded from analysis: (1) coexistent unilateral or bilateral vertebral artery stenosis or luminal irregularity of >50% on MRA or HR-MRI; (2) ruptured dissecting aneurysm associated with subarachnoid hemorrhage; (3) other vasculopathy such as inflammatory arteritis or moyamoya disease; or (4) incomplete follow-up MR examination after the initial HR-VWI.

All patients were treated with dual antiplatelet therapies using aspirin 100 mg and clopidogrel 75 mg daily for six months after symptom onset. If there were no symptoms, the drug was discontinued, and one patient with occlusion of VA continued the medication for six months.

Each patient’s cardiovascular risk factors, including hypertension, diabetes mellitus, and hyperlipidemia, were reviewed. Medical or clinical history, the National Institutes of Health Stroke Scale (NIHSS) score at admission, and drug use history for statins and antiplatelet agents were obtained from medical records.

### 2.2. Magnetic Resonance Imaging Protocol

Our MRI protocol was similar to that of a previous study [11]. A 3.0T MRI scanner (Achieva; Philips Medical Systems, Amsterdam, Netherlands) with a 16-channel head coil was used for MRI. All study subjects initially underwent conventional brain MRI, including three-dimensional (3D) TOF-MRA.

The HR-VWI protocol comprised multiple sequences: black blood (BB) T1-weighted; BB T2-weighted; TOF axial; magnetization-prepared rapid acquisition with gradient echo (MPRAGE); simultaneous non-contrast angiography and intraplaque hemorrhage (SNAP); and contrast-enhanced BB T1-weighted imaging. We used an improved motion-sensitized driven-equilibrium (iMSDE) method to suppress enhanced signals in blood vessels [12,13]. T1-weighted imaging was obtained using a two-dimensional turbo spin-echo sequence under the following conditions: repetition time/echo time, 800/10 ms; field of view, 140 × 140 mm; matrix size, 140 × 150; slice thickness, 2.0 mm; echo train length, 10; and number of excitations, 2. T2-weighted HR-VWI scans also used a turbo spin-echo sequence with the following conditions: repetition time/echo time, 3100/80 ms; field of view, 140 × 140 mm; matrix size, 140 × 140; slice thickness, 2.0 mm; echo train length, 20; and number of excitations, 2. Imaging parameters for TOF-MRA scanning were as follows: repetition time/echo time, 18/3.8 ms; flip angle, 16°; field of view, 140 × 140 mm; matrix size, 312 × 165; slice thickness, 1.0 mm; echo train length, 1; and number of excitations, 3. For 3D MPRAGE sequencing, the segment was acquired using sequence repetition time, inversion preparation time, and phase encoding order obtained from MPRAGE sequences. Image parameter settings were as follows: repetition time/echo time/inversion time, 8.8/5.3/304 ms; flip angle, 15°; echo train length, 32; field of view, 140 × 140 mm; and matrix, 216 × 198. SNAP imaging examined the coronal section from the carotid artery to the VBA. The parameters of the SNAP sequence have been described previously [14]. Image parameters were as follows: TR/TE/TI = 10/4.7/490 ms, FA = 11°, ETL = 98, FOV = 149 × 149 mm, matrix = 187 × 216, scan time = 3 m 30s. The BB technique based on pre-regional 80 mm thick saturation pulses, which saturated incoming arterial flow, was used for all scans. Gadoterate meglumine (0.1 mmol/kg body weight; Dotarem; Guerbet, Aulnay-sous-Bois, France) was administered intravenously to all patients before contrast-enhanced BB T1-weighted imaging. Contrast-enhanced BB T1-weighted imaging was performed approximately five minutes after contrast injection.

The longitudinal coverage of each artery of BB T1- and T2-weighted imaging was 22–24 mm. Each scan lasted 3–4 min. The total scan time was up to 35 min.

### 2.3. Image Analysis

This study examined the presence of VBAD across all samples using HR-VWI. Between them, two neuroradiologists with 26 and 25 years of experience, respectively, and expert vessel-wall-imaging radiologists performed all these procedures.

We defined VBAD as wall thickening or aneurysmal dilatation with low signal intensity due to an intimal flap, a double lumen sign on BB T2-weighted imaging, and high signal intensity due to an intimal flap on TOF-MRA [11]. This study defined a mural hematoma as the detection of the arterial wall’s eccentric, intermediate-to-high signal intensity according to hemorrhagic age using MPRAGE/SNAP [15]. To distinguish a dissection flap from an inflow artifact usually found at the center of the lumen, we considered a layer that extended to the arterial sidewall on any serial image as a dissection flap [15]. Serial follow-up HR-VWI examinations were performed to investigate changes in the double lumen, intramural hematomas, wall enhancement, and lumen dilatation or narrowing. The vessel ratio was defined as maximal dilated vessel size/proximal normal vessel size. Therefore, we classified subjects into three groups based on serial follow-up HR-VWI: type 1 = mural wall thickness of dissected artery during follow-up MR imaging; type 2 = no interval changes of intimal flap or vessel wall; and type 3 = occlusion of a target vessel. We also analyzed serial changes in the wall thickening of the dissected artery.

## 3. Results

Of the 31 patients with VBAD on initial HR-VWI, 16 were excluded because of an incomplete follow-up study (two with aneurysmal rupture, six with patient’s refusal, and eight with limited follow-up study). Fifteen patients (median age of 50 years, nine males) with initial and serial HR-VWI during the follow-up period were enrolled in this study. The demographic data of all patients are shown in Table 1. Thirteen (86.6%) patients complained of headaches at admission. Ten (66.7%) patients had a history of drinking alcohol, and 11 (73.3%) had a dissection in the VA. Also, seven (43.8%) patients had hypertension.

Imaging findings from the initial and final follow-up HR-VWI examinations are shown in Table 2. All patients showed an intimal flap and a double lumen on initial HR-VWI. Twelve (80%) patients showed strong wall enhancement, and nine (60%) had an intramural hematoma. Four (26.7%) patients had severe stenosis related to dissection and two (13.35) showed occlusion of the lumen. The median vessel ratio between a normal vessel and a maximally dilated vessel was 1.9 (range, 1.3–9.0).

In the final follow-up HR-VWI examination, one patient showed complete occlusion of the dissected artery. We analyzed imaging findings from follow-up HR-VWI examinations for 14 patients. Of these patients, nine (60.0%) showed disappearance of the intimal flap. Two patients showed an intramural hematoma in the final follow-up MR imaging. Of 11 patients with initial strong enhancement of the intimal flap, four (26.7%) showed this enhancement at the final HR-VWI examination. Of three patients with initial occlusion, two showed luminal patency. The median vessel ratio in final follow-up imaging was 1.3 (range, 0.8–3.4).

Finally, nine (60.0%) patients showed type 1 lesions due to attachment of the intimal flap to the vessel wall (Figure 1), five (33.3%) showed type 2 lesions (Figure 2), and one showed a type 3 lesion (Figure 3). Patients with type 1 lesions did not show the intimal flap. Three months later, the first follow-up MR imaging showed wall thickening of the dissected artery in all patients with type 1 lesions. Of nine patients with type 1 lesions, four (44.4%) showed complete normalization of vessel lumen on follow-up MR imaging one year later. Patients with type 2 lesions did not show any changes in initial imaging findings for VBADs during the follow-up period. All four patients with BA dissection showed type 2 lesions without changes in the intimal flap or the double lumen. For one patient with an initial dissected aneurysm (type 2 lesion), dilatation of the vessel lumen appeared to be increased in follow-up HR-VWI.

## 4. Discussion

For most patients with VBAD and ischemic symptoms, serial follow-up using HR-VWI showed focal nodular lesions in the vessel lumen with the disappearance of the intimal flap, intramural hematoma, or stenosis seen on initial examination.

DSA, TOF-MRA, and CT angiography (CTA) are conventionally used as the diagnosis and follow-up methods for arterial dissection. Imaging findings reported include VA stenosis (51%); string and pearls (48%); arterial dilatation (37%); arterial occlusion (36%); and pseudoaneurysm, double lumen, and intimal flap (22% each) [16]. In cases where conventional angiography was the reference standard, CTA was more sensitive than MRA or Doppler ultrasonography [16]. Ahn et al. [1] reported that 74% of the dilatation-without-stenosis group showed no change in imaging findings, whereas improvement was observed in 91% of patients with stenosis without dilatation. VBADs with intramural hematoma showed improvement in 63% of cases, with progression occurring in 20% of cases. However, they used TOF-MAR for evaluating intramural hematoma without analysis of the intimal flap or the double lumen.

HR-VWI had higher agreement with the final diagnosis and better inter-rater reliability than the DSA for evaluating VBADs [15,17]. In VWI, true and false lumen by the intimal flap could be visualized as BB [3,18]. Intramural hematoma also becomes hyperintense between a few days and 2 months on T1-weighted imaging, with signal intensity showing changes with time [19,20]. Our study used the iMSDE method for BB imaging and SNAP/MPRAGE sequence for dissecting intramural hematomas. An excellent BB technique and optimization sequence for detecting intramural hematoma shows good diagnostic performance for diagnosing VBAD [21,22,23]. In particular, 3D-SNAP sequences were more sensitive for diagnosing intramural hematoma with higher signal intensity of intramural hematoma than T1-weighted sequences [23]. In our study, all (100%) patients had the intimal flap and the double lumen on VWI, 80% (12/15) showed a strong enhancement of the intimal flap, and 60% (9/15) showed intramural hematoma. The presence of the intimal flap in all patients enrolled in this study may indicate that this is a common feature of VBDAs. Hashimoto et al. [24] have reported that the spontaneous cure rate is 100% for cases in which the signal of intramural hematoma changes over time. However, the spontaneous cure rate for cases without signal change was only 23%. Post-contrast VWI showed enhancement of the vessel wall and the intimal flap of the dissected artery, although the enhancement mechanism was unclear [10,18]. Therefore, a VWI method that can directly show the inner and outer vascular walls, intramural hematomas, and vessel lumen such as the intimal flap and the double lumen is useful for VBADs.

Ono et al. [1] have reported that most patients with symptomatic intracranial arterial dissection and initial hemorrhage show early recurrence with a mean interval of 4.8 days. However, cases in our study had only ischemic symptoms and/or headaches. We did not include ruptured dissecting aneurysms. Recently, it has been shown that ruptured dissecting should not be used for endovascular treatment due to the risk of rebleeding.

Kwon et al. [4] have reported structural changes between intracranial and extracranial dissections of initial and follow-up HR-VWI examinations. Stenosis improvement, disappearance of wall enhancement, and intramural hematoma were more frequent in an extracranial dissection than in an intracranial dissection. However, the median follow-up interval of the intracranial dissection group was 40.0 days. This interval was shorter than that of the extracranial dissection group. Lee et al. [25] have reported intracranial arterial dissection using HR-MRI with variable follow-up periods. Only six patients underwent follow-up HR-MRI after 12 months. A double lumen did not appear in any patients. In addition, most patients’ intramural hematomas and intimal flaps disappeared [25].

In our study, serial HR-VWI was performed after symptom onset for at least 12 months. At the 12-month follow-up HR-VWI examination, one patient showed complete occlusion of the dissected vessel due to compensatory flow of a large contralateral VA. Of the remaining 14 patients, two showed intramural hematoma on final follow-up MR imaging, and nine (60.0%) showed disappearance of the intimal flap. These results were similar to those of a previous study [25]. In addition, four patients showed complete normalization of the dissected lumen. In our study, all patients who showed disappearance of the intimal flap in follow-up HR-VWI examinations had type 1 lesions with changed wall thickness of the dissected artery. Thus, changed wall thickness might be an atrophic change of the intimal flap during the follow-up period. Five patients with type 2 lesions did not show any changes from their initial imaging findings of VBADs.

Our study has several limitations. First, this was a single-center study with only 15 patients enrolled retrospectively. Thus, selection bias might have occurred. Further studies with larger sample sizes are needed to confirm these findings and assess the long-term follow-up outcomes of VBADs. Second, VBADs were diagnosed based on radiological findings from HR-VWI without DSA because we thought that HR-VWI was a useful diagnostic tool for evaluating vessel lumen and imaging findings such as the intimal flap, the double lumen, and intramural hematomas in patients with intracranial dissection.

## 5. Conclusions

Most patients with VBADs showed an excellent healing process during the follow-up period. The intimal flap of dissected VBADs showed changed wall thickness due to atrophic changes in follow-up HR-VWI examinations. This suggests that conservative management approaches may be effective in treating VBADs in patients with ischemia and/or headache and that HR-VWI is not only beneficial for the correct diagnosis of VBADs but also helpful in monitoring VBADs over time, potentially reducing the need for endovascular treatments.

## Figures and Tables

**Figure 1 diagnostics-13-03585-f001:**
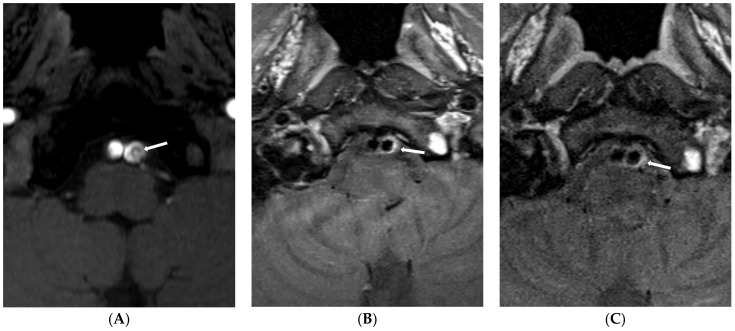
Type 1 lesion in a 41-year-old man (case 13) with mental changes. (**A**) A source image of time-of-flight showing intimal flap and double lumen in the left distal VA (arrow); (**B**) Follow-up T1-weighted imaging after six months showing eccentric wall thickening in the previous dissected portion (arrow); (**C**) Follow-up T1-weighted imaging one year later showing decreased eccentric wall thickening (arrow).

**Figure 2 diagnostics-13-03585-f002:**
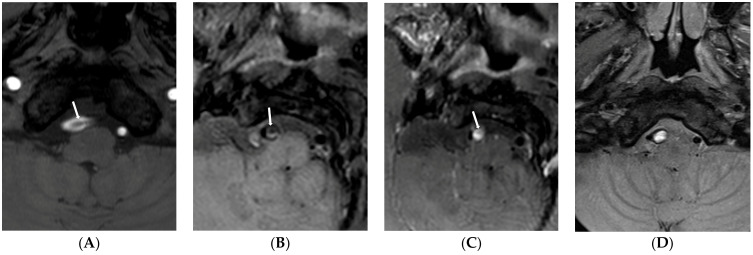
Type 2 lesion in a 67-year-old man (case 6) with initial headache and anopsia. (**A**) A source image of time-of-flight showing intimal flap and double lumen in the right distal VA (arrow); (**B**) Initial T1-weighted imaging showing intimal flap and dilated dissecting aneurysms (arrow); (**C**) Initial contrast-enhanced T1-weighted imaging showing strong enhancement of false lumen (arrow). This finding suggests a T1-contrast effect by contrast stagnation in the false lumen; (**D**) Follow-up T1-weighted imaging after 48 months showing no change of dissecting aneurysms with continuous high signal intensity in the false lumen.

**Figure 3 diagnostics-13-03585-f003:**
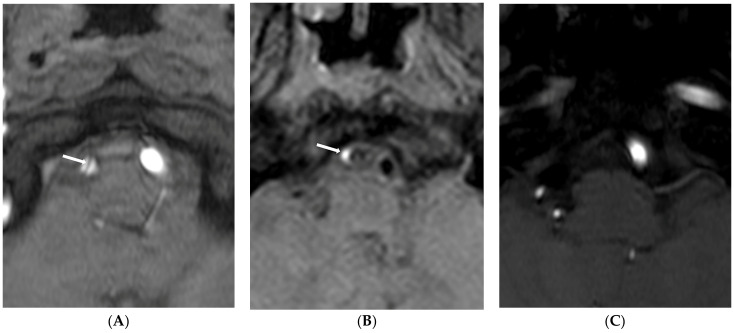
Type 3 lesion in a 44-year-old woman (case 1) with initial headache. (**A**) A source image of time-of-flight showing intimal flap and double lumen in the right distal VA (arrow); (**B**) Initial T1-weighted imaging showing eccentric high-wall intensity in the false lumen (arrow); (**C**) Follow-up source image of time-of-flight one year later showing no visualization of the right VA.

**Table 1 diagnostics-13-03585-t001:** Demographic data of all patients.

	Age	Sex	Symptoms	Systolic Pressure	Diastolic Pressure	Glucose Level	Cholesterol	LDL	Smoking	Alcohol	Location	Total F/U (Months)
1	44	F	Headache, neck pain	126	90	94	200	131	0	1	VA	12
2	45	F	Dizziness	125	76	110	145	89	0	0	VA	12
3	64	F	Headache, neck pain	177	67	120	170	110	0	1	VA	12
4	41	M	Headache	166	100	99	291	230	0	1	VA	12
5	41	F	Headache, dizziness	130	79	111	158	95	1	1	VA	12
6	67	M	Headache, anopsia	163	83	285	151	117	0	1	VA	48
7	52	M	Headache	191	120	86	173	100	0	1	VA	18
8	50	F	Headache	117	53	89	207	136	0	0	VA	12
9	76	M	Headache	125	86	105	135	69	0	0	BA	12
10	48	M	Headache, orbital pain	130	80	104	194	108	0	1	VA	12
11	57	M	Headache, neck pain	120	68	105	192	125	1	1	BA	12
12	40	M	Headache, neck pain	145	89	120	209	160	0	0	VA	18
13	41	M	Mental change, headache	141	101	115	163	111	1	1	VA	41
14	51	M	Dizziness	178	99	109	187	119	1	1	BA	12
15	79	F	Headache, neck pain	105	58	105	165	109	0	0	BA	24

Note: VA, vertebral artery; BA, basilar artery; F/U, follow-up; LDL, low-density lipoprotein; M, male; F, female.

**Table 2 diagnostics-13-03585-t002:** Imaging findings from initial and final follow-up HR-VWI examinations.

	Initial MR Examination						Final MR Examination						Type	Final Wall Findings
	Intimal Flap	IMH	CE	Stenosis	Occlusion	Vessel Ratio	Intimal Flap	IMH	CE	Stenosis	Occlusion	Vessel Ratio		
1	1	1	1	0	1	1.6	0	0	0	0	1		3	Occlusion
2	1	1	1	1	0	2.0	0	0	0	0	0	1.0	1	Normalization
3	1	1	1	1	0	2.6	0	0	1	1	0	0.8	1	Wall thickness
4	1	0	1	0	0	1.9	0	0	0	0	0	1.0	1	Normalization
5	1	1	0	0	1	5.0	0	0	0	0	0	1.0	1	Wall thickness
6	1	1	1	0	0	2.3	1	1	0	0	0	2.9	2	Increased dissecting lumen
7	1	1	1	1	0	2.4	0	0	1	0	0	2.0	1	Wall thickness
8	1	1	1	1	0	1.9	0	0	0	1	0	0.9	1	Normalization
9	1	0	1	0	0	1.9	1	0	0	0	0	1.9	2	Intimal flap
10	1	0	1	0	0	2.0	0	0	1	0	0	1.3	1	Normalization
11	1	0	1	0	0	1.8	1	0	0	0	0	1.8	2	Intimal flap
12	1	1	1	0	0	1.4	0	0	1	0	0	1.0	1	Wall thickness
13	1	1	1	0	0	3.6	0	0	0	0	0	3.4	1	Wall thickness
14	1	0	0	0	0	1.4	0	1	0	0	0	1.4	2	Intimal flap
15	1	0	0	0	0	1.3	1	0	0	0	0	1.3	2	Intimal flap

Note: IMH, intramural hematoma; CE, contrast enhancement.

## Data Availability

The data presented in this study are available upon request from the corresponding author. The data are not publicly available due to privacy restrictions.

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
