# Peer review of "Changes in Vertebrobasilar Artery Dissection Visible with High-Resolution Vessel Wall Imaging: A Serial Follow-Up Study"

_diagnostics, 2023, doi:10.3390/diagnostics13233585_

Round 1

Reviewer 1 Report (Previous Reviewer 1)

Comments and Suggestions for Authors

The article presents high resolution vessel wall imaging follow-up of vertebral arteries dissection. This is a relatively rare pathology, difficult to diagnose. The cohort of patients is significant and follow-up, both clinical and imagistic, is consistent. The patterns identified for the imaging evolution of vessel wall bring important data, relevant for the therapeutic approach. It appears that a significant part of the patients have a good evolution, with healing of the dissection. 

Author Response

Thanks

Reviewer 2 Report (Previous Reviewer 2)

Comments and Suggestions for Authors

Thank you for answering all my questions. In this version, you correct all points that I questioned. However,  I find it difficult to accept that you only made changes in the text without further explanation and reliable evidence.  Especially, the correction about the selection of the research subjects and number of cases analyzed.

In table 1, what is the blood pressure level, blood sugur level, lipid level. 

Do all 16 patients receive the same dual antiplatelet therapy? 

Comments on the Quality of English Language

Moderate editing of English language required

Author Response

In table 1, what is the blood pressure level, blood sugur level, lipid level. 

Do all 16 patients receive the same dual antiplatelet therapy? 

 Answer) All patients underwent the medical treatment as same protocol.

Comments on the Quality of English Language

Moderate editing of English language required

  • We performed the English editing from professional editing team.

Round 2

Reviewer 2 Report (Previous Reviewer 2)

Comments and Suggestions for Authors

I have no more comment.

This manuscript is a resubmission of an earlier submission. The following is a list of the peer review reports and author responses from that submission.

Round 1

Reviewer 1 Report

Comments and Suggestions for Authors

The article addresses the question of vertebral dissection follow-up.

There is little knowledge of the vertebral dissection evolution, the disease is rare. The results in this paper offer high-resolution imaging insights on vertebral wall healing after dissection. The design of the study is appropriate, correct follow-up of patients with vertebral dissection. Imaging supports the conclusions. References are appropriate and discussed inside the paper. Figures offer relevant images on the described features of evolution after vertebral dissection.

Very interesting paper. Good design of the study. High quality imaging provided. Important results in term of therapeutic implications, in a rare pathology.

Reviewer 2 Report

Comments and Suggestions for Authors

This article described the High-Resolution Vessel Wall Imaging of 16 patients with Vertebrobasilar Artery Dissection, and observed the changing in 24m follow-up.

Comments

1 In the abstract, subjects were mentioned as a retrospectively enrolled, but in the method section, the description of enroll process is unclear.

2 L146-149, 33 patients were evaluated initially, 16 were excluded, 15 were enrolled. What happen the 2 remain?

3 during 24m follow-up period, what treatment were applied to those 16 enrolled patients? Dose the treatment affect the prognosis of this disease.

4 in tab. 1 the main parameter of hypertension, diabetis, smoking, hyper[1]lipidemia and alchol should be reported, rather than just record with or without.

Comments on the Quality of English Language

Quality of English Language is good.